# Value judgment of new medical treatments: Societal and patient perspectives to inform priority setting in The Netherlands

**Anna Nicolet** *, **Antoinette D. I. van Asselt**, **Karin M. Vermeulen**, **Paul F. M. Krabbe**

Department of Epidemiology and Health Systems, Institute of Social and Preventive Medicine (Unisanté), Lausanne, Switzerland

* anna.nicolet@unisante.ch

## Abstract

### Background

In many countries, medical interventions are reimbursed on the basis of recommendations made by advisory boards and committees that apply multiple criteria in their assessment procedures. Given the diversity of these criteria, it is difficult to find common ground to determine what information is required for setting priorities.

### Objective

To investigate whether society and patients share the same interests and views concerning healthcare priorities.

### Methods

We applied a framework of discrete choice models in which respondents were presented with judgmental tasks to elicit their preferences. They were asked to choose between two hypothetical scenarios of patients receiving a new treatment. The scenarios graphically presented treatment outcomes and patient characteristics. Responses were collected through an online survey administered among respondents from the general population (N = 1,253) and patients (N = 1,389) and were analyzed using conditional logit and mixed logit models.

### Results

The respondents' preferences regarding new medical treatments revealed that they attached the most relative importance to additional survival years, age at treatment, initial health condition, and the cause of disease. Minor differences in the relative importance assigned to three criteria: age at treatment, initial health, and cause of disease were found between the general population and patient samples. Health scenarios in which patients had higher initial health-related quality of life (i.e., a lower burden of disease) were favored over those in which patients' initial health-related quality of life was lower.

**Data Availability Statement:** All data (Stata format) is available at the public repository Figshare, under doi 10.6084/m9.figshare.10000556.

**Funding:** Financial support for this study was provided entirely by ZonMw (The Netherlands Organization for Health Research and Development; grant number 152002053). The funding agreement ensured the authors' independence in designing the study, interpreting the data, writing, and publishing the report.

**Competing interests:** The authors have declared that no competing interests exist.

## Conclusions

Overall, respondents within the general population expressed preferences that were similar to those of the patients. Therefore, priority-setting studies that are based on the perspectives of the general population may be useful for informing decisions on reimbursement and other types of priority-setting processes in health care. Incorporating the preferences of the general population may simultaneously increase public acceptance of these decisions.

## 1. Background

Governments in many Western countries rely on the recommendations of advisory boards and committees when making decisions on the reimbursement of new drugs. Decisions to reimburse non-pharmaceutical treatments are also increasingly based on assessments of the available evidence by these independent parties to determine whether the technology in question offers added value to patients and society at large. In the USA, the emphasis is on comparative effectiveness research, which entails direct comparisons of healthcare interventions to determine which ones work best for which patients and which ones pose beneficial or harmful outcomes [1]. In England and Wales, the National Institute for Health and Care Excellence advises the National Health Service on the clinical relevance and cost-effectiveness of treatments, whose health effects are expressed in quality-adjusted life years (QALYs) [2]. Elsewhere in Western Europe, assessment procedures used for this purpose are diverse. In this paper, we will elaborate on the assessment procedure used in one Western European country, the Netherlands [3–5].

In the Netherlands, the Appraisal Committee (Advies Commissie Pakket) of the National Health Care Institute (Zorginstituut Nederland) provides advice on whether or not to include certain care services or treatments in the basic insurance package. The main criteria used to assess the therapeutic and societal values of drugs and other health interventions are necessity, efficacy, cost-effectiveness, and feasibility. However, some of the sub-criteria can also have significant impacts on the final decision regarding the reimbursement of health interventions [6]. For example, despite low cost-effectiveness, reimbursement may be considered when no other treatment is available, when an orphan disease is in question, or when the burden of disease is substantial. The Appraisal Committee also takes account of societal value judgments in their deliberations. These judgments are not based solely on clinical relevance and cost-effectiveness; equity issues and moral values are also considered. For instance, the fair allocation and distribution of medical care is considered an important aspect of social and distributive justice.

Given this diversity of criteria, it is difficult to find common ground for determining what information is required for setting priorities. While a number of studies have explored the views of the general public on the principles that should guide priority setting [7–9], others have investigated the criteria that are actually used in priority setting [10, 11]. Tanios et al. [11] found a convergence among decision makers regarding the relevance of the criteria that were considered. However, in general, there seems to be a substantial degree of plurality of opinions regarding the identified criteria. Accordingly, Van Exel et al. [7] concluded that it is unlikely that a single decision rule could satisfactorily cover all of the relevant equity principles and viewpoints.

Another concern relates to the question of whose values should be applied for priority setting: those of the experts, the general public, or the patients. Most advisory boards and

committees tasked with decision making on reimbursement use criteria that are derived from experts' opinions. When health effects are expressed in QALYs, this usually implies that the applied values are derived from representative community samples [12]. As taxpayers, members of the general public are assumed to adhere to principles of justice and equity as opposed to self-interest. However, it is also often argued that patients are the best judges of their own health situations, as they are likely to be better informed than healthy people or more adept at visualizing certain health states [13–17]. To the best of our knowledge, head-to-head studies aimed at eliciting views on priority setting within general populations as opposed to patients' views have not been conducted so far.

In summary, a clear set of criteria to guide priority setting is lacking for the Netherlands, and the views of the general public or patients are not explicitly considered in the decision-making process. One of the questions emerging from this situation is whether societal interests and views concerning healthcare priorities accord with those of patients, who are most affected by them. Therefore, the aim of this study was to investigate whether the societal perspective on healthcare priorities reflects the interests and views of patients.

## 2. Methods

### 2.1 The discrete choice methodology and criteria selection

Discrete choice experiments (DCEs) are widely used to elicit personal and societal preferences in health valuation studies [18, 19]. DCEs have also been applied to inform a wide range of health policy, planning, and resource allocation decisions within various healthcare settings. The statistical literature classifies DCEs within the framework of probabilistic discrete choice models [20–25]. All DCE models establish the relative merit of a phenomenon based on its relative attractiveness. This method requires participants to choose from among two or more scenarios (choice tasks) described using specific attributes with distinct levels. The respondents were required to make complex priority choices for the study's aim to be achieved. Accordingly, we decided to use the DCE method, as it simulates the choice-making process in a real-world setting and can therefore be considered less cognitively demanding. However, when using this method, the selection and identification of the most important and informative attributes (and their levels) must be performed carefully to enable respondents to process these attributes without becoming fatigued. In the present study, the attributes included in the paired scenarios represented criteria. They were carefully selected to ensure that the essential aspects of the decision-making process were captured.

In light of our review of the available studies, we decided to present the relevant health outcomes in a way that enabled them to be easily understood. The conventional QALY concept served as a starting point for developing this approach. We created a graphical representation of a QALY (including levels and durations of particular health states) that was extended to incorporate other relevant and important criteria. To ensure that our selection of criteria (attributes in the DCE) was appropriate and well-informed, we performed an extensive review of the literature (publications from 2014 onward) using the Medline and Embase databases. The review was conducted with the aim of extracting a set of criteria that reflected societal concerns about treatment effectiveness and equity considerations [26].

Presenting all of the criteria (n = 25) within a single choice task would have made the task too demanding for the respondents, and many of the criteria were only applicable in very specific circumstances. Therefore, each of the 25 criteria was individually assessed by two authors and one senior researcher within the research team, and differences in their assessments were discussed to reach a consensus. Finally, a combination of essential and prominent criteria that are receiving attention within the Dutch (and international) social value appraisal system were

selected. Apart from reflecting the relevant treatment outcomes, the criteria also reflected the characteristics of the potential recipients of the new treatment to enable an assessment of its necessity according to the burden and cause of a disease. The scenarios were constructed on the basis of the following factors:

a. Relevant outcomes of new and standard treatments: any changes in a patient's health-related quality of life (HRQoL) after undergoing a new treatment or after a standard treatment (if it exists and is accessible) and any gains in life years for a patient after undergoing a new treatment or a standard treatment (if it exists and is accessible).

b. Relevant characteristics of the patient: age, the initial HRQoL of a patient burdened by a disease or by an exacerbated health condition, and the cause of an acute onset (deterioration of the current HRQoL) associated with an accident, genetics, or an unhealthy lifestyle. The definition of disease or exacerbated health condition includes injury; illness; a handicap; genetic deficit; or another physical, mental, or nervous condition, disorder or ailment.

We decided to focus on acute onsets rather than on chronic or reversible conditions. The distinction of levels of the attributes in a DCE necessitate step-wise changes in HRQoL, whereas chronic conditions are commonly characterized by a trajectory of slow deterioration, often without any clear episodes. Therefore, acute onsets are the only conceivable cause of sudden and dramatic differences in HRQoL. We applied the term "acute" to denote the sudden onset of a new disease, the sudden deterioration of an existing one, or the occurrence of an accident.

## 2.2 Scenarios

Each scenario covered a health condition that existed prior to the acute onset, the effect of the standard treatment, and the effect of a new treatment. There have been numerous debates regarding the added benefits of using graphical representations to enhance task comprehension in DCEs [27–31]. Specifically, it has been suggested that diagrams and other graphical representations could facilitate respondents in making complex choices among health states in DCEs conducted online [31]. Although not all attributes can be presented graphically, the most important attributes (length of life, quality of health/HRQoL, and additional gains or losses) can be depicted in this way. Therefore, instead of using conventional textual descriptions in the DCE, we opted for graphical representations of the scenarios, which were expected to enhance respondents' comprehension. To reduce possible framing bias relating to the use of graphs and icons to describe hypothetical states, we inserted written explanations and notes to support the graphical presentation. This combination of graphic and textual material was used because the findings of previous studies indicate that graphical representation helps respondents to understand tasks better, whereas written explanations facilitate their judgment [30]. The DCE was designed to ensure that none of the scenarios were implausible or unrealistic. Face-to-face pilot testing of the survey was performed at the University of Groningen as well as online in April 2015. There was no time limitation set for the pilot testing, although the mean duration of the face-to-face version, including instructions and feedback, was estimated to be 20 minutes. The online version was typically shorter.

During the pilot phase, the following questions were verbally posed to the respondents (n = 8): Were the tasks easy? Was the presentation attractive and comprehensible? Were you able to make a choice? The respondents indicated that the task was not difficult to understand, and they confirmed that the graphic design facilitated comprehension of the task. The respondents also provided some suggestions for improving the instructions provided with the task. Values and assumptions derived from the existing body of literature [32–37] were used for

portraying the initial health state (which could include an underlying disease, or not), the cause of the acute onset, health gains after the patient commenced the new treatment, and health gains after the patient commenced the standard treatment (Table 1). A detailed explanation of the values and assumptions used for the level selection is presented in S1 Appendix and S1 Fig.

Because the task of comparing options on the basis of several criteria is a demanding one, we restricted the number of criteria used to construct the scenarios. Only the most relevant ones were included so as not to overload the respondents with information.

### 2.3 Choice tasks

The respondents were provided with an explanation of the paired scenarios along with instructions on how to proceed with the task. Combinations of criteria were presented in three steps: the patient's initial health state, followed by the effects of the new treatment, and lastly a comparison of the effects of the new and standard treatments. The patient's age and HRQoL before the acute onset were respectively depicted on the x- and y-axis of the graph, and the benefits of the new and standard treatments were color-coded and depicted as shaded areas on a plane. The cause of the acute onset was depicted by two icons, one of which represented unhealthy lifestyle elements, such as smoking/being overweight, and the other represented external factors, such as a genetic predisposition or an accident. To reduce the cognitive burden on respondents, a three-step approach was implemented that facilitated their understanding of the distinct attributes in the scenarios. As they progressed through the three consecutive steps, the respondents gradually gained familiarity with each attribute before they finally encountered the actual choice task. The first step (see Fig 1A) presented a hypothetical patient's age and initial health state, depicting this individual's HRQoL before the occurrence of the acute event. The second step introduced new information, namely gains from the new treatment in terms of additional life years and post-treatment HRQoL (Fig 1B). Only when they reached the third step (Fig 1C) did the respondents encounter the final scenario comprising the patient's health state before the onset of the acute event, the benefits of the new treatment, and the benefits of the standard treatment, if this information was available. The respondents then had to decide which of the two scenarios they preferred for the hypothetical patients according to the information available to them. The question associated with the task was framed as follows: "Which patient should receive the new treatment?" Additionally, arrows and balloons

**Table 1. Attributes and their corresponding levels used to describe criteria in the scenarios.**

| Attributes | | Levels |
|---|---|---|
| Initial health state | | |
| | Age at onset | 25, 50, 75 |
| | Initial HRQoL | 0.5, 0.7, 0.9 |
| Effects of new treatment | | |
| | Change in HRQoL | -0.2, -0.1, 0.0 |
| | Gain in LY | 2, 10, 20 |
| Effects of standard treatment | | |
| | Change in HRQoL | -0.2 |
| | Gain in LY | 0*, 2, 10, 20 |
| Cause of acute onset | | unhealthy lifestyle (1), accident/genetics (0) |

*In case a standard treatment is not available

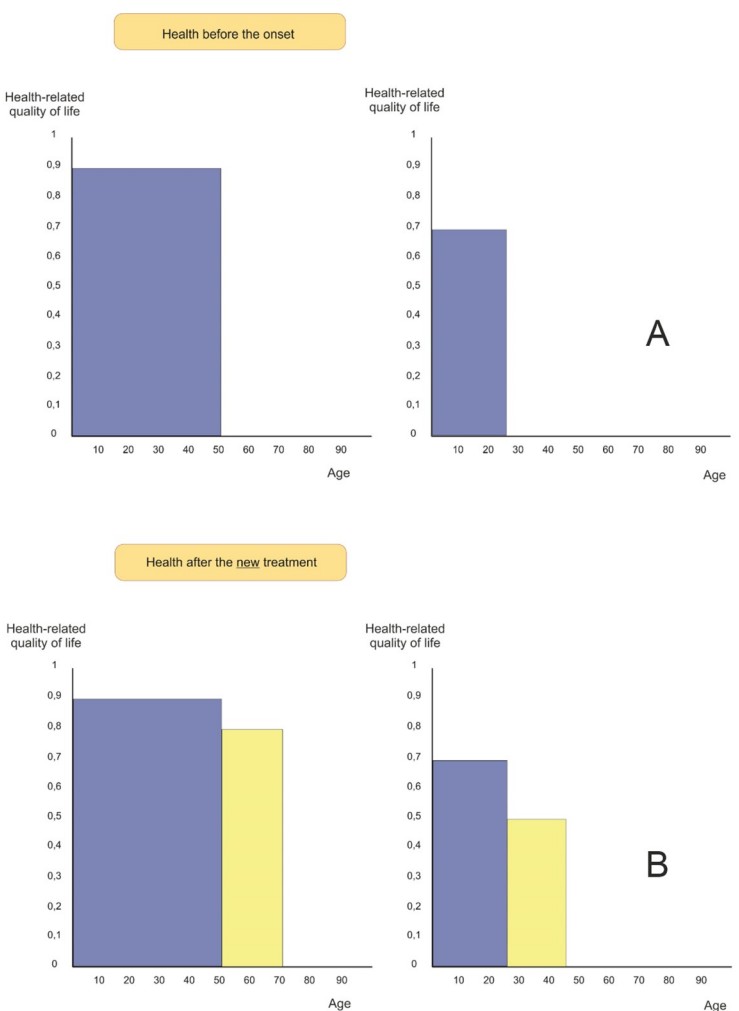

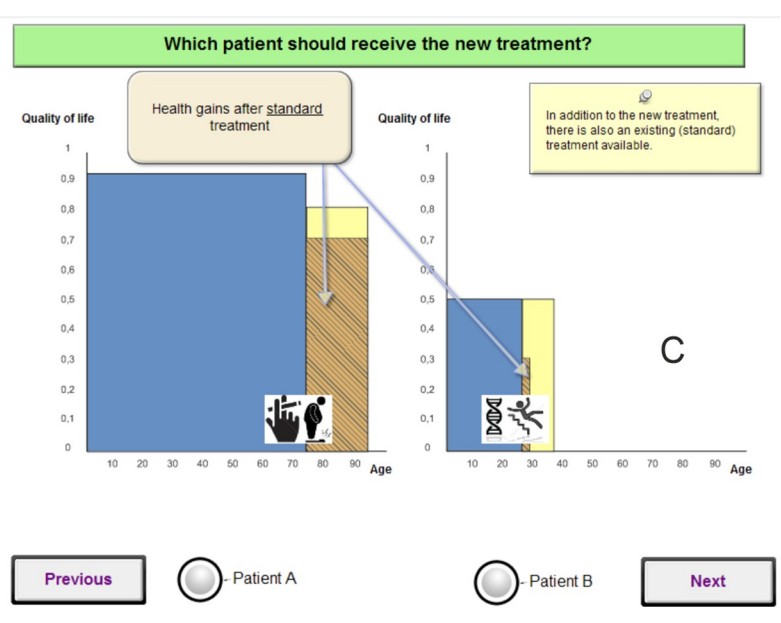

**Fig 1. Example of the three-step discrete choice response task.** a) Step 1. Health before the onset of the acute event. b) Step 2. Health after the onset of the acute event and the administration of the new treatment. c) Step 3. Actual choice task. Health after the acute event and the administration of the new treatment and standard treatment and the cause of the acute onset.

were introduced to explain where the gains from the standard treatment or the new treatment could be found on the screen.

## 2.4 Study design

We created a total of 200 paired scenarios. To diminish the burden on respondents and to avoid fatigue, this set was subdivided into 20 blocks, each comprising 10 choice tasks. The tasks were selected from a range of possible combinations of paired comparisons using an efficient design (the Ngene software program, the MNL model, and null priors) [18]. Additional constraints were applied (see Appendix A) to impose the new treatment's superiority, which essentially implied that compared with the standard treatment, the new treatment would always be associated with an equal or higher HRQoL. The survey was designed as a web-based experiment. The respondents were randomly assigned to one of the 20 blocks so that each respondent completed only 10 response tasks.

## 2.5 Respondents

A sample drawn from the Dutch population and from a panel of patients who were at least 18 years old were contacted by a market research agency Survey Sampling International [SSI] based in Rotterdam, the Netherlands. Individuals from the patient panel were recruited and requested to self-report their diagnosis. We defined patients as individuals with current diseases or serious complaints. Therefore, we did not consider individuals with past experience of a disease. The rewards for participation, which are typically vouchers or electronic gift cards, were arranged vis-à-vis internal agreements between the participants and SSI. The patients were asked to indicate whether they had any of the following disease types: diabetes, neck and back problems, heart disease, hearing or vision loss, asthma/COPD, eczema, mental health problems, stroke, rheumatism, cancer, epilepsy, lung disease, or gastrointestinal disease. Patients who had more than one diagnosed disease were allowed to report multiple diseases. The sampling design for the general population did not enable us to verify whether potential respondents had been diagnosed with a disease. The Medical Ethics Review Committee at the University Medical Center Groningen issued a waiver for this study, stating that the pertinent Dutch legislation (the Medical Research Involving Human Subjects Act) was not applicable to this non-interventional survey study (METc 2014.181).

## 2.6 Analysis

The data were analyzed using the McFadden conditional logit model [38, 39] with dummy-coded variables representing the levels of the attribute (Stata, clogit routine). The probit model (Stata, asmprobit routine) relaxing the independence of irrelevant alternatives has been executed but was outperformed by the conditional logit model (goodness-of-fit results are not presented). A mixed logit model (mixlogit) that accounted for random variations in preferences across respondents (heterogeneity) was also considered. However, we determined that the basic conditional logit model was sufficient for meeting the aims of the present experimental study, which focuses on overall preferences.

A large number of respondents were required to obtain precise estimates for the (paired) scenarios, According to Lancsar and Louviere [40], 20 respondents per survey block (20 blocks

in this study) are sufficient to ensure that a model is reliable, but a bigger sample is required to conduct a significant post-hoc analysis. The minimal number of respondents for the present study was 400 patients and 400 members of the general population, but a larger sample was recruited to allow the possibility of conducting a further analysis.

We conducted three comparative analyses of the general population and the patients. First, we graphically depicted the regression coefficients for the criteria and their levels to convey relative positive or negative preferences. Second, we applied the range method to compare the relative importance assigned to the criteria for each sample [41]. We calculated the range between the coefficients for the individual levels, which were then converted into proportions. We applied this method to calculate and compare the difference in the preference weights for the best and worst levels of a criterion. This difference yielded an estimate of the relative importance of that criterion over the range of levels.

$$W_{\text{attribute(i)}} = \frac{\max Ci - \min Ci}{\sum_j (\max Cj - \min Cj)}$$

Third, we conducted an analysis using the asclogit routine in Stata that included second-order interactions between respondent types and all of the criteria in the joint sample. The statistical significance of an interaction indicated that the general population and the patients had different preferences regarding the criterion of interest.

## 3. Results

### 3.1 Respondents

Data were collected between July 2015 and January 2016. A total of 1,986 respondents from the general population and 2,256 patients were invited to participate in the study. Sociodemographic information provided by SSI indicated that with reference to age and sex, both of the recruited samples (from the general population and the list of patients) were representative of the Dutch population. Because of privacy restrictions, the background characteristics were collected only at the stage the participants were invited, and were subsequently aggregated. Therefore, this information was not available for the subset of respondents who were included in the analysis after completing the survey. Some patients had been diagnosed with more than one disease, and the most common diagnoses were neck and back pain, diabetes, and asthma/COPD. Out of all of the individuals who registered for the survey (Table 2), the numbers of respondents who completed the survey, and whose answers were included into the analysis, were 1,253 (63% of invited persons) for the general population sample and 1,389 (62% of invited persons) for the patient sample. We performed a quality check of the responses by identifying respondents who consistently chose either the left or right options as their responses. Only 36 respondents (0.01%) demonstrated this response pattern. Therefore, we decided to retain these respondents in the main analysis, as such a modest proportion would not bias the main results. Additionally, because the study design did not include any clearly dominated scenarios, we could not rule out any kind of pattern as being invalid.

### 3.2 The relative importance of the criteria

Table 3 shows the parameter estimates for the conditional logit model, and Fig 2 shows the attribute weights. The results indicating the relative importance of the attributes and their weights obtained using the alternative mixed logit model did not differ substantially from those obtained using the conditional logit model. These results are presented in S1 Table. For example, respondents from both the general population and patient samples indicated that a

**Table 2. Characteristics of the participants in two sub-samples used in the study.**

| Characteristics | General population Overall registered N = 1,986 | Patients Overall registered N = 2,256 |
|---|---|---|
| Female, N (%) | 1104 (56) | 1239 (55) |
| Age, mean (SD) | 46.6 (14.4) | 47.8 (14.0) |
| Age group, N (%) | | |
| 18–24 | 286 (14) | 244 (11) |
| 25–34 | 194 (10) | 223 (10) |
| 35–44 | 240 (13) | 281 (12) |
| 45–54 | 485 (24) | 580 (26) |
| Older 55 | 781 (39) | 928 (41) |
| Diagnosed with*, N (%) | | |
| Neck and back pain | - | 995 (44) |
| Diabetes | - | 736 (33) |
| Asthma/COPD | - | 418 (19) |
| Mental health problems | - | 383 (17) |
| Hearing or vision loss | - | 370 (16) |
| Eczema | - | 352 (16) |
| Rheumatism | - | 335 (15) |
| Heart disease | - | 302 (13) |
| Gastrointestinal disease | - | 168 (7) |
| Cancer | - | 153 (7) |
| Lung disease | - | 86 (4) |
| Stroke | - | 80 (4) |
| Epilepsy | - | 53 (2) |

*The total frequencies exceeded 2,256 because some patients were diagnosed with more than one disease.

gain in life years following the new treatment had the most significant effect (weights of 0.31 vs. 0.35 in the conditional logit model and 0.3 vs. 0.34 in the mixed logit model). The criteria ranked second for the general population respondents were age and cause of disease (similarly, 0.22). Respondents strongly favored treating a 25-year-old over a 50-year-old, and this effect was even stronger for a 75-year-old patient. Respondents in the patient group rated the cause of the disease as the second most important criterion (0.23), which was only slightly higher than their ranking of age at acute onset (0.22). The initial HRQoL (0.14 for the general population sample vs. 0.12 for the patient sample) and the treatment's effect of maintaining the HRQoL at the same level as it was prior to the acute onset (0.08 for the general population sample vs. 0.07 for the patient sample) were the least important criteria within both groups. The outcomes of the standard treatment had no significant effect (0.03 for the general population sample vs. 0.02 for the patient sample).

### 3.3 Differences between the general population and patient samples

Overall, the results for the relative importance of the criteria were almost identical for the two samples. However, there were statistically significant differences relating to preferences for specific criteria within the total sample. This analysis revealed statistically significant second-order interactions for combinations of the sample type with the following criteria: age, initial HRQoL (0.9), cause of acute onset, change in HRQoL, and life years gained by the respondent after undergoing the new treatment (S2 Table). The significance of the two criteria (age and cause of the disease), as revealed in this analysis, supported the results of the analyses

conducted for each individual sample (Table 3). Finally, although the interactions of the sample type with the change in HRQoL and the initial HRQoL were statistically significant, they did not lead to any modification in our interpretation of the relative importance of the attribute, as these criteria remained among those that were ranked the lowest by respondents in both samples.

## 4. Discussion

This study, which was aimed at investigating whether society and patients share the same interests and views concerning healthcare priorities, was operationalized using two samples of respondents who expressed their preferences for a specific treatment in a specific situation,

**Table 3. Parameter estimates (clogit) of the six criteria for the two sub-samples (based on completed surveys).**

| | General population (SE) | Patients (SE) |
|---|---|---|
| | N = 1,253 | N = 1,389 |
| | Obs = 47,756 | Obs = 51,932 |
| **SCENARIO CRITERIA** | | |
| **Patient characteristics** | | |
| *Age* | | |
| Age 25 (reference) | - | - |
| Age 50 | -0.17 (0.03)* | -0.10 (0.03)* |
| Age 75 | -0.67 (0.04)* | -0.52 (0.04)* |
| *Initial Health-Related Quality of Life (HRQoL)* | | |
| HRQoL 0.5 (reference) | - | - |
| HRQoL 0.7 | 0.28 (0.03)* | 0.24 (0.03)* |
| HRQoL 0.9 | 0.42 (0.03)* | 0.28 (0.03)* |
| *Cause of acute onset* | | |
| Accident, genetics (reference) | - | - |
| Unhealthy lifestyle | -0.65 (0.03)* | -0.55 (0.03)* |
| **New treatment outcomes** | | |
| *HRQoL change after new treatment ($\Delta$HRQoL)* | | |
| $\Delta$HRQoL -0.2 (reference) | - | - |
| $\Delta$HRQoL -0.1 | 0.16 (0.03)* | 0.05 (0.03) |
| $\Delta$HRQoL 0 | 0.25 (0.03)* | 0.17 (0.03)* |
| *Life years gained after new treatment ($LY_{new}$)* | | |
| $LY_{new}$ 2(reference) | - | - |
| $LY_{new}$ 10 | 0.64 (0.04)* | 0.55 (0.04)* |
| $LY_{new}$ 20 | 0.94(0.04)* | 0.84 (0.04)* |
| **Standard treatment outcomes**\*\*\* | | |
| *Life years gained after standard treatment ($LY_{standard}$)* | | |
| Standard treatment unavailable (reference) | - | - |
| $LY_{standard}$ 2 | -0.04 (0.03) | 0.00 (0.03) |
| $LY_{standard}$ 10 | -0.04 (0.03) | 0.02 (0.03) |
| $LY_{standard}$ 20 | -0.10 (0.05)** | 0.01 (0.04) |
| Goodness-of-fit | -14679 | -16445 |

\*P < 0.01

\*\*P < 0.05

\*\*\* the change in Health-Related Quality of Life (HRQoL) after administering the standard treatment had no variance, as it was a fixed attribute with one possible level of -0.2 (the 7th criterion)

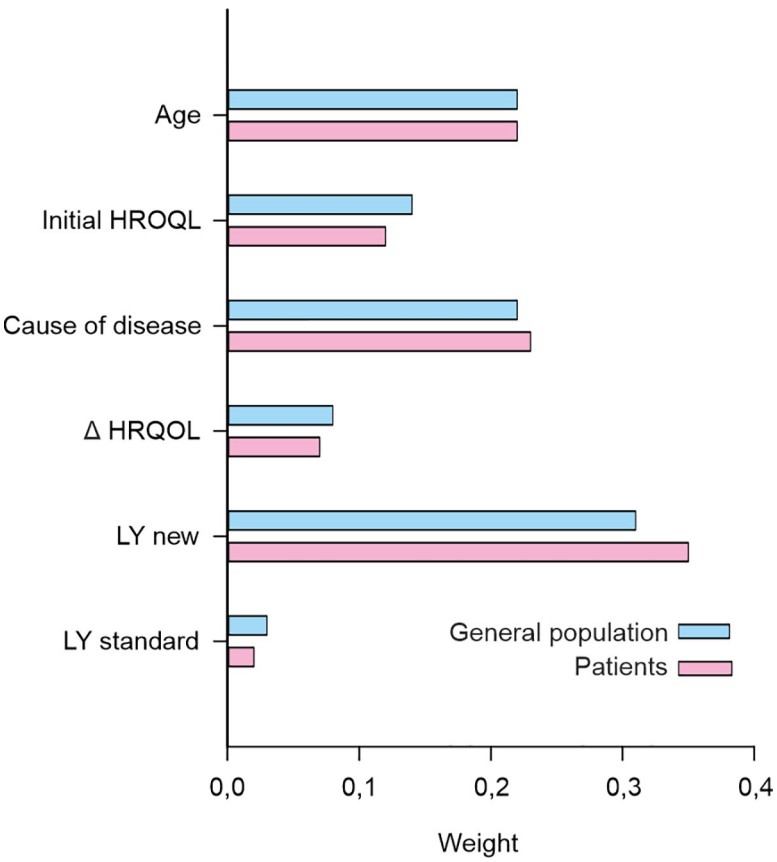

**Fig 2. Attribute weights for the general population group and the patient group.**

defined by particular criteria. These criteria, comprising the age of the patient and the effectiveness of the treatment, were expected to capture societal concerns, especially those relating to treatment effectiveness and equity considerations.

The results of the analysis revealed the relative importance of additional survival years attributed to the new treatment. Similar results were reported in a recent study [37], which showed that respondents tend to favor scenarios in which QALY gains are significant. The results of the present study showed that preferences related to new treatments also depend on patient characteristics (such as the individual's age or initial health state). In a German study, criteria for prioritizing health care based on patients' personal characteristics were examined. The findings of this study indicated that disease severity and patients' HRQoL are the most important attributes, whereas an unhealthy lifestyle was ranked lowest [42]. However, these researchers did not associate an unhealthy lifestyle with the cause of a disease, as we did in the present study. Our findings as well as those of Skedgel et al. [37] suggest that preferences do not strictly and exclusively follow QALY-maximizing decision rules; rather, they incorporate the characteristics of both the patient and the treatment. The findings of another study [43] also indicate that age (i.e., children or a younger population) and saving lives, even when HRQoL values are low, are deemed important. The findings of the present study partly support those of the latter study, which emphasized the importance of saving younger patients. The availability and effect of a standard treatment appeared to have no effect on the appraisal of new treatments. This finding concurs with that of Green et al. [19], who found that the availability of "other treatments" was the least important attribute.

Our results can be considered in the context of Williams' "fair innings" argument [44]. According to this argument, every individual is entitled to a lifespan that is considered reasonable or "fair." The "fair innings" argument takes account of the characteristics of both the patient and the treatment (in terms of health gains) [45, 46] over this individual's entire life span. Thus it takes account not only of the patients' past health and actual ages but also of their future health. According to Williams [44], life years gained for individuals who have not had a "fair innings" should be valued above life years gained for people whose "innings" have been fair. However, our findings did not endorse this argument. We observed that age and a better initial health status (i.e., health in the past and actual age) were accorded a high level of societal importance along with health gains experienced after the new treatment (i.e., future health). The findings of two studies conducted, respectively, in Canada [37] and Sheffield (in the UK) [47] also indicated a strong preference for providing treatment to younger individuals. However, in a study conducted in the UK, Lancsar [48] found that while preference weights for the patient's age at acute onset were small, weights for the patient's age at death were larger. Evidently, the importance of age is not solely attributable to the fair innings principle; other reasons for favoring younger patients could relate to their economic or social productivity.

The argument concerning the "burden" or "severity of disease" has been widely applied within the literature on societal preferences [8, 46, 49, 50]. Shah [49], who conducted a literature review, observed that an individual's pre-treatment health state is the most widely used criterion for defining the burden of disease. We therefore incorporated this definition as initial HRQoL into our study. There is considerable heterogeneity in existing definitions of the burden of disease and in the associated study methods (personal trade-off, DCE, and social welfare function), which may influence a study's outcome. Nevertheless, Shah pointed out that in most studies, respondents were, on the whole, inclined to prioritize the severely ill. Skedgel et al. [47] also found that severely ill patients were favored over those who were less severely ill. This finding does not accord with our own findings, which showed that those patients whose initial HRQoL levels were higher were favored over those whose initial HRQoL levels were lower. There are two possible explanations for this finding. The first explanation could be that the respondents were inclined to favor younger and generally healthier individuals because they are economically and demographically productive. This strategy would therefore contribute the most to overall economic well-being within society. A second possible explanation relates to the HRQoL loss of 0.2 after the acute event that was depicted in the graph and how it was visually perceived. A level of HRQoL that is initially high would still appear high, even after a loss of 0.2 or 0.1 (in a large area of the graph), compared with an initially low HRQoL, which could worsen or remain at a low level (in a smaller area of the graph). Similar results were reported by Wetering et al. [51], demonstrating higher preferences for treating individuals who are already in a relatively good state of health before undergoing treatment. In that study, the researchers used graphical representations of the scenarios in which specific areas showing losses were depicted.

Our findings also demonstrated the importance of a lifestyle-related cause; a criterion that is rarely taken into consideration. The argument about individuals who are responsible for the causes of their diseases was raised in an earlier study by Singh et al. [52], which emphasized that the public accorded higher priority to interventions for diseases in which the patient has no control over the cause of the disease. Conversely, they accorded lower priority to programs for treating illnesses that were "self-inflicted." In another study focusing on the prioritization of investments in health service innovations [53], the researchers reported that the respondents were negatively inclined toward funding innovations that targeted individuals with drug addiction and obesity issues. Although the respondents in our study tended to choose the alternatives associated with more survival years after the new treatment had been administered,

they prioritized younger patients, those whose acute onset stemmed from an accident or genetic cause, and those with a higher initial HRQoL. This finding accords with those of other studies conducted in various countries that focused on societal preferences [37, 54, 55]. For example, the findings of Luyten et al. [54], who conducted a study in Belgium, indicate a stronger preference for individuals who did not cause their own illnesses. The findings of Gu et al. [8] suggest that in general, the young are favored over the old, the more severely ill are favored over the less severely ill, and individuals with self-induced illnesses tend to receive lower priority than those whose illnesses are not self-induced. In studies in which health gains were considered, larger gains were universally preferred. These findings trigger the interest to explore the interactions between lifestyles and various other criteria.

Our findings demonstrated that patients and members of the general population are able to compare different health outcomes and to express their preferences for one particular outcome. In most countries, including the Netherlands, the task of prioritizing healthcare interventions is mainly delegated to experts. To strengthen public acceptance and consensus on how healthcare interventions should be prioritized, a nation-wide assessment of public preferences, such as the one reported in our study, could be considered. Our finding that differences in the preferences of patients and non-patients are modest is an important one, as it suggests that assessments could be harmonized relatively easily.

Within the literature, there is no consensus regarding the issue of whether values relating to healthcare priorities differ between patients and members of general population. For instance, dissimilarities in the values for health states between these samples were reported in a number of previous studies [30, 55–58]. Although we did not find any substantial differences between the general population and patients, the analysis of specific interaction terms did reveal a difference between the two samples relating to the relative importance of criteria (the cause of the acute onset was more important to patients than to members of the general population). The patients prioritized individuals whose diseases were induced by accidents or by genetic predispositions, indicating that patients may attach more importance to individuals who take responsibility for their own health. However, this observation could be attributed to the icon format used to depict the cause of the disease.

In the present study, we assumed that a graphical format would add value because it is attractive and relatively easy to comprehend. We also assumed that the use of supplementary visual aids, such as pop-up notes and balloons, would improve the respondents' understanding. However, a graphical representation could plausibly influence the respondents to some extent. For example, the design could differentially influence the decisions of respondents who focus on the sizes of the graph areas compared with those who focus more on instructions. Moreover, the icons could be perceived differently when presented together with a graph or a diagram. As colorful and easy-to-understand elements, icons may initially capture the attention of respondents. Consequently, respondents could pay more attention to an attribute that is presented as an icon than to the same attribute presented in a conventional textual format. Therefore, it would be useful to provide respondents with step-by-step visual guidance and verbal or even video-recorded instructions that effectively discourage them from ignoring any of the elements on the screen.

During the pilot-testing phase, we confirmed the ease of comprehension and the appeal of our study design. However, providing the respondents with an additional opportunity to leave their feedback about the survey design and the challenges they faced could enhance the quality of the responses. These additional inputs would contribute to a deeper understanding of how the respondents executed the choice tasks and the decision strategies that they used.

This study had a few limitations that should be noted. First, although the two samples were representative of the Dutch population in terms of sex and age, the respondents from the

general population were, on average, slightly younger than the patients. Moreover, the number of individuals below the age of 45 years within the general population sample was higher than the number of such individuals within the patient sample. It is plausible that younger respondents within the general population sample would favor scenarios in which younger patients were presented.

Second, the sampling design for the general population did not allow for verification of whether potential respondents had been diagnosed with a disease. Although we acknowledge that the general population includes patients and that some of the individuals may have had a disease at the time of their participation in the survey, our aim was not to investigate representative proportions of healthy and non-healthy individuals within the general population. Therefore, the general population was considered to comprise a mix of individuals with and without diseases and not just healthy individuals. However, access to information on the health states of respondents from the general population would enable the performance of subgroup analysis that would yield additional information about the heterogeneity of preferences for new treatments in the Netherlands. In general, these results should be interpreted with caution, as the representativeness of the samples in terms of the respondents' health states, education levels, and socioeconomic statuses were not considered. Only age and sex were taken into account when determining the representativeness of the samples.

Third, our decision to focus on the acute onset of non-reversible deterioration of health conditions limits the generalizability of the results. Although many diseases are characterized by non-reversible deterioration of the quality of a patient's health, for some diseases or individuals, there could be opportunities for patients to return (partially) to their initial health levels. Our DCE design limited the representation of such outcomes and thus neglected a proportion of actual health trajectories that can be observed in practice. Finally, we acknowledge the importance of qualitative research in which members of the general population or patients are involved in the selection of attributes and their levels, although the current study did not include this type of research. As the current study implies a high level of awareness and knowledge of priority assessment processes in the Netherlands, patients and the general population were not involved in this conceptual stage of criteria selection. However, members of the general population were involved in a subsequent pilot testing phase and were able to express their opinions regarding the comprehensibility of the tasks.

In conclusion, we found that five out of the six presented criteria affected respondents' preferences for specific treatment scenarios, constituting a multifaceted system of distinctive and even conflicting arguments and criteria. Nevertheless, our study demonstrated that both patients and members of the general population were able to make choices regarding the prioritization of healthcare interventions. Moreover, patients and members of the general population demonstrated similar patterns of preferences, indicating that the preferences of the general population could be representative of the views of patients in the area of priority setting relating to health care. The assessment process may be enhanced through the involvement of members of the general population in priority assessments, in addition to experts, thereby strengthening public acceptance of reimbursement and other decisions relating to healthcare priority setting.

## Supporting information

**S1 Fig. Graphical explanation.** Explanation of various options in the scenarios: (A) Possible health states before onset; (B) Example of new treatment for 25-year-old patient with 0.9 HRQoL; (C) Standard treatment for 25-year-old patient with 0.9 HRQoL (if accessible and

exist).
(TIF)

**S1 Table. Parameter estimates (mixed logit) of the 6 criteria for the two sub-samples (based on completed surveys).**
(DOCX)

**S2 Table. Parameter estimates of the 6 criteria with interaction terms (sample type × criteria) for the combined sample of general population and patients.**
(DOCX)

**S1 Appendix.**
(DOCX)

**S2 Appendix.**
(PDF)

## Author Contributions

**Conceptualization:** Anna Nicolet, Antoinette D. I. van Asselt, Karin M. Vermeulen, Paul F. M. Krabbe.

**Data curation:** Anna Nicolet.

**Formal analysis:** Anna Nicolet, Paul F. M. Krabbe.

**Funding acquisition:** Paul F. M. Krabbe.

**Investigation:** Anna Nicolet, Antoinette D. I. van Asselt, Karin M. Vermeulen, Paul F. M. Krabbe.

**Methodology:** Anna Nicolet, Paul F. M. Krabbe.

**Project administration:** Paul F. M. Krabbe.

**Resources:** Paul F. M. Krabbe.

**Software:** Anna Nicolet, Paul F. M. Krabbe.

**Supervision:** Paul F. M. Krabbe.

**Validation:** Anna Nicolet, Antoinette D. I. van Asselt, Karin M. Vermeulen, Paul F. M. Krabbe.

**Visualization:** Anna Nicolet, Paul F. M. Krabbe.

**Writing – original draft:** Anna Nicolet, Antoinette D. I. van Asselt, Karin M. Vermeulen, Paul F. M. Krabbe.

**Writing – review & editing:** Anna Nicolet, Antoinette D. I. van Asselt, Karin M. Vermeulen, Paul F. M. Krabbe.

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
