## [Decision Letter · Decision Letter 0]

20 Feb 2020

PONE-D-19-29090

VALUE JUDGMENT OF NEW MEDICAL TREATMENTS: SOCIETAL AND PATIENT PERSPECTIVES

PLOS ONE

Dear Dr Nicolet,

Thank you for submitting your manuscript to PLOS ONE. After careful consideration, we feel that it has merit but does not fully meet PLOS ONE’s publication criteria as it currently stands. Therefore, we invite you to submit a revised version of the manuscript that addresses the points raised during the review process.

Addressing the 3 reviewers' comments will likely require substantial revisions to the manuscript.  Please ensure that adequate information about the development of the DCE is included in the revised manuscript to be able to assess the validity and reliability of the approach. If necessary, consider including an appendix with additional details. Please ensure that the study data are available as per journal policy.  

We would appreciate receiving your revised manuscript by Apr 05 2020 11:59PM. To enhance the reproducibility of your results, we recommend that if applicable you deposit your laboratory protocols in protocols.io, where a protocol can be assigned its own identifier (DOI) such that it can be cited independently in the future. For instructions see: http://journals.plos.org/plosone/s/submission-guidelines#loc-laboratory-protocols

We look forward to receiving your revised manuscript.

Kind regards,

Jan Ostermann

Academic Editor

PLOS ONE

Journal Requirements:

Reviewers' comments:

Reviewer's Responses to Questions

**Comments to the Author**

1. Is the manuscript technically sound, and do the data support the conclusions?

Reviewer #1: Yes

Reviewer #2: Yes

Reviewer #3: Partly

2. Has the statistical analysis been performed appropriately and rigorously? 

Reviewer #1: Yes

Reviewer #2: I Don't Know

Reviewer #3: Yes

3. Have the authors made all data underlying the findings in their manuscript fully available?

Reviewer #1: No

Reviewer #2: Yes

Reviewer #3: Yes

4. Is the manuscript presented in an intelligible fashion and written in standard English?

Reviewer #1: No

Reviewer #2: Yes

Reviewer #3: Yes

5. Review Comments to the Author

Reviewer #1: This article deals with a very important topic and challenging question, namely how reimbursement procedures could be improved by incorporating societal and/or patient needs more explicitly in reimbursement criteria. The answer to this question needs to be addressed via conducting research such as the study here presented. However, the following points of critique should be discussed/incorporated:

" The title, methods of abstract and methods section needs to mention more explicitly that the study was conducted in the Netherlands/with Dutch participants. The discussion should include a discussion of how these findings (do not) translate to other countries in Europe/US, in view of the nationally organized reimbursement system as pointed out in the introduction

" Title: consider adding: VALUE JUDGMENT OF NEW MEDICAL TREATMENTS: SOCIETAL AND PATIENT PERSPECTIVES to inform priority setting in the Netherlands

" Abstract methods: include how many participants (patients and general population separately), type of survey (online?), type of analysis

" Abstract conclusion should include a stronger recommendation; what is the key take-away for priority setting that could be recommended based on this research?

" Consider changing "discrete choice model" to "discrete choice experiment" (DCE), as this term is more commonly used

" Methods section in abstract and paper: ensure this is written in past tense, eg in methods of abstract "The scenarios graphically represented treatment outcomes and patient characteristics"

" Methods section:

o More information necessary about the selection/development of criteria/attributes; who and how was this done exactly? Qualitative research (e.g. interviews, focus groups such as the nominal group technique) with the stakeholder group eventually answering the DCE survey is recommended for the selection of DCE attributes. Was qualitative research performed? Were patients/participants of the general public involved for the selection? To what extent was this driven by currently used criteria for reimbursement (i.e. top down approach) vs bottom-up (e.g. via patients/general public)? Another method for limiting the criteria would have been via grouping them. These aspects (rationale, importance of qualitative research) should be discussed/reflected on both in the methods section (why did you choose a certain approach) and any limitations as well as reflections on the different ways for attribute selection and the impact of these different ways on the final results should be discussed in the discussion section.

o Why DCE and not another elicitation method such as swing weighting, best worst scaling? The rationale for DCE should be in the methods and any potential impact of this choice on the results in the discussion

o Why a focus on acute onsets?

o P.5: delete "rigorous". Instead more information on the methods for this literature review should be included (databases consulted, year of literature review)

o P. 7 delete "representative", as this was only tested for the sex and age of participants

o Clarify what "SSI" means

o What types of rewards were given to participants?

o Pilot: how long did it take for participants to fill it in?

" Results:

o More detail necessary what is meant with "substantially"

o was there open comment field/question where participants could give feedback on the survey; how easy/difficult, areas for improvement/relevance of the characteristics/whether the characteristics captured the most important factors influencing their preference? If so this should be discussed in the results. If not, this should be discussed in the discussion

" Discussion:

o More reflection on methodological choices and potential limitations:

" See above for comment regarding attribute selection

" Sample has several potential limitations, representativeness was only tested for sex and age as other characteristics were not captured. This should be discussed in the limitations

" From the general population there is no certainty that they weren't ill at the time of participation since this information was not gathered; this limitation should be clearly discussed in the discussion section and it should be clearly stated in the methods that the general population may have included ill participants

" "our decision to focus on acute onset of non-reversible deterioration of health conditions limits the generalizability of the results": explain in more detail in what way the generalizability is limited

o Discuss more extensively: What does this research learn us moving forward for the priority setting procedure in the Netherlands (and potentially beyond) Should patients or citizens be more actively involved eg via DCE surveys? Any critique on the current process and how this should be approached moving forward, based upon this study? You identified the most influential criteria, how should this result be incorporated in priority setting?

" Language: the article could benefit from a review by a native English speaker as there are some typos, e.g. "A Germany study" should be "a study in Germany"

Reviewer #2: This manuscript describes an original research, designed to investigate whether society and patients have the same interests and views concerning healthcare priorities.

Both the study process and the rationale behind each detail is very well described and explained in detail. The method chosen by the researchers, in which hypothetical scenarios were presented to the participants graphically with written explanations and notes is remarkable.

Overall, the article is very interesting and written extremely well.

A few points should be noted regarding the criteria and results:

1. The meaning of the notion "initial HRQoL" in the study is not clear. On p. 5 of the manuscript, the authors explain that the criterion of initial HRQoL had been chosen to reflect burden of disease. However, from the explanation in S1 Appendix, it is understood that this notion relates to the patient's state before the acute onset, while the burden of disease can be reflected only through the gap between the patient's state before and after the acute onset?! It may be assumed that the preference of patients with higher initial HRQoL was affected by this failure.

2. No wonder that the outcomes of the standard treatment had no significant effect; the relevant effect for the decision to choose a new treatment is the difference between the gains of the new treatment and the standard one. So, the outcomes of the standard treatment would be expected to have an effect on the participants' choice only as long as they are significantly worse than the outcomes of the new treatment. This argument is somewhat reflected in the results, through the opposite correlation between the gains of the standard treatment and the effect related to it by both groups of participants.

Reviewer #3: Review Comments to the Author

Major Comments

My main quibble with this paper is the way that the scenarios and choice tasks are presented in the paper almost incidentally – i.e. without proper justification (for readers), explanation of how the approach works (given it is novel) and consideration of how participants found using the approach.

To me at least, the graphical, symbolic presentation of the scenarios is quite unorthodox – and potentially useful … How was this new approach developed? What theoretical and practical considerations and trade-offs were confronted in order to settle on the final design? How did participants find using the approach? Did any of them experience difficulties? Did others like the approach? Overall, is the method valid and reliable? What improvements (refinements) might be made for future uses of the method? (Should the method be used again?)

The best we are told in terms of justification for the approach is: “Instead of conventional descriptions written in text, the scenarios consisted of graphical representations, which are presumably easier to comprehend and may reduce certain types of bias [27]. To reduce possible framing bias by using graphs and icons to describe hypothetical states, we added written explanations and notes to support the graphical presentation. Such a combination was used because previous research suggested that graphical representation helps the respondent to understand the task better, while written explanations facilitate judgment [64].”

“PRESUMABLY easier to comprehend and MAY reduce certain types of bias”?! Based on just one reference? And where is reference 64?

Are there any drawbacks of using graphs and icons? (It seems obvious to me that there are!) Are there any concerns about participants’ interpretations (surely, meanings and interpretations of images/icons is a notoriously tricky area)? We are not told how you settled on the designs you used … Did you test alternative presentations?

In a nutshell, my concerns in this regard are twofold: (1) Does your graphical and icon-based approach work for participants (such that their preference data are valid and reliable)?; and (2) Will readers be able to understand the approach as you have explained it in the paper?

With respect to question #1, I could find little evidence – reassurance – in the paper that participants understood the approach. (Even if some/many of them did not, that would still be an interesting results, especially what you concluded from such a finding.) The survey did not seem to ask participants what I would have thought are two obvious questions: “How easy / difficult did you find answering the choice questions?” “How confident are you in your answers?” The answers to these questions would be interesting. They might also shed some light on your results with respect to the relative importance of the attributes.

There seems to be no ‘data quality controls’ at all – such as the time taken to answer the questions, consistency checks (repeated questions; or the inclusion of dominated ones), counts of participants clicking the same answer (left or right patient) every time. Overall, how do you know that you have ‘good quality’ data? (Does it matter?)

When I looked at Figure 1, I was initially bamboozled as to what I was seeing – and so I would not have been able to express a valid choice between the two patients. (Are the images in the software dynamic? On page 7 we are told that “three steps” were involved. Maybe the approach is obvious when working on a computer screen?) After re-reading section 2.2 and 2.3 several times and puzzling over Figure 1, I finally got it! (Phew!) However, I have to say that, when I finally understood what I was being asked, I would have no confidence in my answers to questions like that (but perhaps I am unusually stupid – it’s possible :) ).

(Is it possible that requiring participants to, first, notice and understand the six criteria represented in Figure 1, and then cognitively process them and the associated trade-offs, is asking too much of most people? Might the cognitive burdenbe too much? Apart from the eight pilot-testers, how would you know?)

In passing, in the Discussion, you muse, “However, it can be hypothesized that the graphical representation might influence the respondents to some extent. For example, the design might affect the decisions of respondents who focus on the sizes of graph areas differently than respondents who focus more on instructions.” But that seems to be it!

With regards to question #2, you need to explain to your readers how your approach works, in particular, how to interpret Figure 1. (To be honest, I am a bit surprised you have not received this feedback earlier – even from yourselves as readers – and made improvements.) It seems to me – and you can tell by how much I am discussing it here! – that your approach for eliciting people’s preferences is central to the paper. And so if readers can’t understand the approach, then the paper pretty much fails, in my opinion. (Sorry!)

Other Comments

In the second-last paragraph of page 13 you mention a surprising result: “This is not in line with the findings of our study, which revealed that the patients with higher initial HRQoL were favored over those with lower initial HRQoL.” Any ideas why?

At the top of page 14, you say, “respondents did not prefer innovations targeting people with ‘drug addiction’ and ‘obesity’”. Do you mean this? Or instead do you mean, “respondents preferred that innovations did not target people with ‘drug addiction’ and ‘obesity”? (I don’t think these two statements are the same.)

Typos and Stylistic Errors

P. 2: “collected from general population” should be “collected from the general population”.

P. 2: “Preferences were strongly and significantly affected by …” is too vague. Do you mean “the relative importance of the attributes …”?

P. 2: “in the assessment of prioritizing new medical interventions” does not read well.

P. 3: “which consist of direct comparison of health care interventions to determine which work best for which patients and which pose” should be “which consists of direct comparison of health care interventions to determine which works best for which patients and which poses”.

P. 3: “value judgements into their consideration” should be “value judgements in their consideration”.

P. 4: “Summarizing” would be better as “In summary”.

P. 4: Add “for the Netherlands” to the end of the phrase, “Summarizing [In summary], a clear set of criteria to guide priority setting is lacking”.

P. 5: “among two (paired)” is inelegant.

P. 10: “Of the total amount of respondents” is also inelegant. How about just, “Of the respondents”?

P. 11: Perhaps “the importance of the criteria” should be “the relative importance of the criteria”? (And elsewhere.)

P. 13: “gained life years” should be “life years gained”.

P. 14: “These finding triggers” should probably be “These findings trigger”.

P. 14: This sentence does not make sense: “In the present study we assumed that a benefit due to the relative easiness and attractiveness of the graphical format and improving it by visual aids, such as notes and balloons, popping up as additional help.”

Best wishes

6. PLOS authors have the option to publish the peer review history of their article (what does this mean?). If published, this will include your full peer review and any attached files.

Reviewer #1: Yes: Rosanne Janssens

Reviewer #2: Yes: Ofra G. Golan, LLD

Reviewer #3: No

---

## [Author Response · Author response to Decision Letter 0]

30 Apr 2020

We would like to thank you for your thorough review of our article and for pointing out areas that required improvement. We have considered all of your comments and are providing our responses in the documents attached (Response to Reviewers)

---

## [Decision Letter · Decision Letter 1]

22 Jun 2020

VALUE JUDGMENT OF NEW MEDICAL TREATMENTS:

SOCIETAL AND PATIENT PERSPECTIVES TO INFORM PRIORITY SETTING IN THE NETHERLANDS

PONE-D-19-29090R1

Dear Dr. Nicolet,

We’re pleased to inform you that your manuscript has been judged scientifically suitable for publication and will be formally accepted for publication once it meets all outstanding technical requirements.

Kind regards,

Jan Ostermann

Academic Editor

PLOS ONE

Additional Editor Comments:

Congratulations. Please consider making the minor change recommended by Reviewer 1.

Reviewers' comments:

Reviewer's Responses to Questions

**Comments to the Author**

1. If the authors have adequately addressed your comments raised in a previous round of review and you feel that this manuscript is now acceptable for publication, you may indicate that here to bypass the “Comments to the Author” section, enter your conflict of interest statement in the “Confidential to Editor” section, and submit your "Accept" recommendation.

Reviewer #1: All comments have been addressed

Reviewer #2: (No Response)

Reviewer #3: All comments have been addressed

2. Is the manuscript technically sound, and do the data support the conclusions?

Reviewer #1: Yes

Reviewer #2: Yes

Reviewer #3: Yes

3. Has the statistical analysis been performed appropriately and rigorously? 

Reviewer #1: Yes

Reviewer #2: I Don't Know

Reviewer #3: I Don't Know

4. Have the authors made all data underlying the findings in their manuscript fully available?

Reviewer #1: Yes

Reviewer #2: Yes

Reviewer #3: Yes

5. Is the manuscript presented in an intelligible fashion and written in standard English?

Reviewer #1: Yes

Reviewer #2: Yes

Reviewer #3: Yes

6. Review Comments to the Author

Reviewer #1: Congratulations on this revised version and thank you for implementing my comments. Great article. Your findings and conclusions now come out more clearly and you carefully acknowledge the limitations. The conclusion is interesting and important for both policy makers and researchers. Just one minor additional comment: in the abstract methods it is still mentioned "discrete choice model"; consider changing to: "We applied a discrete choice experiment in which respondents were presented with judgmental tasks to elicit their preferences."

Reviewer #2: (No Response)

Reviewer #3: Your manuscript is greatly improved, including the quality of your writing. The paper reads well. Well done.

7. PLOS authors have the option to publish the peer review history of their article (what does this mean?). If published, this will include your full peer review and any attached files.

Reviewer #1: Yes: Rosanne Janssens

Reviewer #2: Yes: Ofra G. Golan, LLD

Reviewer #3: Yes: Paul Hansen

---

## [Editor Report · Acceptance letter]

26 Jun 2020

PONE-D-19-29090R1 

VALUE JUDGMENT OF NEW MEDICAL TREATMENTS:
SOCIETAL AND PATIENT PERSPECTIVES TO INFORM PRIORITY SETTING IN THE NETHERLANDS 

Dear Dr. Nicolet:

I'm pleased to inform you that your manuscript has been deemed suitable for publication in PLOS ONE. Congratulations! Your manuscript is now with our production department. 

Kind regards, 

on behalf of

Dr. Jan Ostermann 

Academic Editor

PLOS ONE